# Asymbiotic mass production of the arbuscular mycorrhizal fungus *Rhizophagus clarus*

Sachiko Tanaka [1,8], Kayo Hashimoto [1,8], Yuuki Kobayashi [1], Koji Yano [1], Taro Maeda[1,2], Hiromu Kameoka [1,3,7], Tatsuhiro Ezawa [4], Katsuharu Saito [5], Kohki Akiyama [3] & Masayoshi Kawaguchi [1,6 ✉]

Arbuscular mycorrhizal (AM) symbiosis is a mutually beneficial interaction between fungi and land plants and promotes global phosphate cycling in terrestrial ecosystems. AM fungi are recognised as obligate symbionts that require root colonisation to complete a life cycle involving the production of propagules, asexual spores. Recently, it has been shown that *Rhizophagus irregularis* can produce infection-competent secondary spores asymbiotically by adding a fatty acid, palmitoleic acid. Furthermore, asymbiotic growth can be supported using myristate as a carbon and energy source for their asymbiotic growth to increase fungal biomass. However, the spore production and the ability of these spores to colonise host roots were still limited compared to the co-culture of the fungus with plant roots. Here we show that a combination of two plant hormones, strigolactone and jasmonate, induces the production of a large number of infection-competent spores in asymbiotic cultures of *Rhizophagus clarus* HR1 in the presence of myristate and organic nitrogen. Inoculation of asymbiotically-generated spores promoted the growth of host plants, as observed for spores produced by symbiotic culture system. Our findings provide a foundation for the elucidation of hormonal control of the fungal life cycle and the development of inoculum production schemes.

[1] Division of Symbiotic Systems, National Institute for Basic Biology, Nishigonaka 38, Myodaiji, Okazaki, Aichi 444-8585, Japan. [2] Institute for Advanced Biosciences, Keio University, Kakuganji 246-2 Mizukami, Tsuruoka, Yamagata 997-0052, Japan. [3] Graduate School of Life and Environmental Sciences, Osaka Prefecture University, 1-1 Gakuen-cho, Nakaku, Sakai, Osaka 599-8531, Japan. [4] Graduate School of Agriculture, Hokkaido University, Kita 9, Nishi 9, Kita-ku, Sapporo, Hokkaido 060-8589, Japan. [5] Faculty of Agriculture, Shinshu University, 8304 Minamiminowa, Nagano 399-4598, Japan. [6] Department of Basic Biology, School of Life Science, Graduate University for Advanced Studies (SOKENDAI), Nishigonaka 38, Myodaiji, Okazaki, Aichi 444-8585, Japan. [7] Present address: Graduate School of Life Sciences, Tohoku University, 2-1-1 Katahira, Aoba-ku, Sendai, Miyagi 980-8577, Japan. [8] These authors contributed equally: Sachiko Tanaka and Kayo Hashimoto. ✉email: masayosi@nibb.ac.jp

Arbuscular mycorrhizal (AM) fungi are ubiquitous symbionts of the majority of terrestrial plant species and can facilitate plant mineral acquisition[1,2]. AM fungi are obligate biotrophic fungi, depending on host-derived carbohydrates, such as sugars and lipids[3–6]. Genome analyses of AM fungi have demonstrated that the lack of key metabolic enzymes is involved in the obligate biotrophy[7–11]. AM fungi have long been considered unculturable without the host. However, co-culture of the AM fungus *Rhizophagus irregularis* and mycorrhiza-helper bacterium *Paenibacillus validus* demonstrated that AM fungi can complete their life cycle in the absence of host plants[12,13]. Recently, it has been showed that fatty acids boost AM fungal growth and sporulation under asymbiotic conditions. Palmitoleic acid asymbiotically induced infection-competent secondary spore formation of *R. irregularis*[14]. Further, myristate initiated the asymbiotic growth of AM fungi and can also serve as a carbon and energy source[15]. These findings would lead to the development of new research tools for AM studies and novel production systems of AM fungal inoculants. At present, the in vitro monoxenic culture with carrot hairy roots is one of the prevalent AM fungal culture methods, in which an average of 8500–9000 *R. irregularis* spores can be obtained from 10–15 parent spores[16]. However, spore production in the asymbiotic culture systems remains lower than those in symbiotic co-cultures. Moreover, spores generated with supplying palmitoleic acid or myristate were smaller than those generated symbiotically[14,15], and their performance as inoculum is largely uncertain.

Traditionally, it has been considered that AM fungi are only capable of taking up inorganic nutrients[1]. Recent studies have challenged this notion by demonstrating that AM fungi promote hyphal growth in patches of organic material and obtain nitrogen released during degradation of organic material, independent of the host plants[17,18]. It has also been shown that AM fungi directly take up recalcitrant and labile forms of organic nitrogen[19].

Not only nutrients but signalling molecules from host plants may be crucial for AM fungal growth and reproduction[20–22]. Some phytohormones show positive effects on interactions between AM fungi and hosts. For example, strigolactones are a major plant-derived signal that induces hyphal branching[23] and elongation[24] and stimulates their mitochondrial activity[25] in the pre-symbiotic stage. Methyl jasmonate (MeJA) was increased during AM fungal colonisation in the roots[26], concurrent with the up-regulation of jasmonic acid biosynthesis genes in plant cortical cells containing arbuscules that are highly branched fungal structures for nutrient exchange[27,28]. The knowledge on the roles of phytohormones in the interactions between AM fungi and plants has been accumulated recently[29], but there is only limited information about the direct effect of phytohormones on AM fungal growth[21] and reproduction.

Here, we showed that two phytohormones, strigolactone and jasmonate, play a crucial role in asymbiotic growth and sporulation of AM fungi. During this study, we found that *R. clarus* grew more vigorously than *R. irregularis*. In the presence of myristate and organic nitrogen, a synthetic strigolactone GR24 stimulated the formation of a large number of secondary spores and the addition of jasmonates led to more mass production of spores and increased spore size. Finally, we confirmed that the asymbiotically generated spores of *R. clarus* can be subcultured and facilitate the growth of Welsh onions via establishing root colonisation.

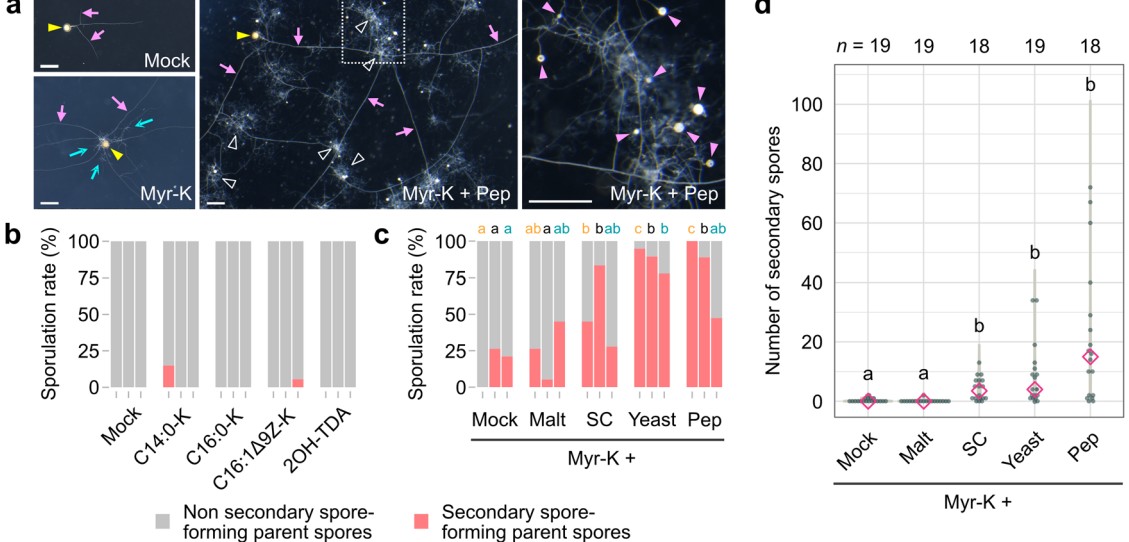

**Fig. 1 Effects of fatty acid salts and organic nitrogen sources on asymbiotic cultures of *R. clarus*. a** Effects of potassium myristate (Myr-K) and peptone (Pep) on *R. clarus* growth. AM fungi were incubated in a medium with or without 100 μM Myr-K and 0.2 g l$^{-1}$ peptone for 6 weeks. Branched hyphae (blue arrows) were observed around a parent spore (yellow arrowhead) in a medium containing Myr-K. Small, white spores were formed among densely packed coils (DPCs, outlined arrowheads) in a medium containing both Myr-K and Pep. The rightmost picture is a magnified image of the dotted box in the middle picture. Pink arrowheads indicate newly generated secondary spores. Pink arrows indicate runner hyphae. Bars = 500 μm. **b** Sporulation rates, percentage of secondary spore-forming parent spores relative to germinated parent spores, in cultures supplemented with 100 μM fatty acid at 6 weeks after incubation (WAI). Three bars in each condition are trials 1–3, respectively, from the left. 2OH-TDA 2-hydroxytetradecanoic acid. C14:0-K potassium myristate. C16:0-K potassium palmitate. C16:1Δ9Z-K potassium palmitoleate. **c** Sporulation rates in cultures with 0.2 g l$^{-1}$ organic nitrogen source in the presence of Myr-K at 6 WAI. Different letters above of the graph indicate significant differences among treatments in each trial using Fisher's exact test with Bonferroni correction ($p < 0.05$). Malt, malt extract. SC, SC dropout. Yeast, yeast extract. **d** Numbers of secondary spores generated from a single parent spore in medium supplemented with each organic nitrogen in the presence of Myr-K at 6 WAI. Diamonds indicate medians. Statistical significance was calculated using the Wilcoxon rank-sum test with Bonferroni correction. Different letters indicate significant differences ($p < 0.05$). $p$-values are described in Supplementary Data 2.

## Results

**Survey of base media containing fatty acids and organic nitrogen sources for asymbiotic culture of *R. clarus* HR1.** We developed a base medium for *R. clarus* HR1 asymbiotic culture. According to genomic analysis of *R. clarus*, this fungus lacks metabolic pathways of disaccharide degradation and thiamine biosynthesis[7]. Thus, we adjusted the composition of the original modified M medium[12] by adding glucose and more thiamine and reducing sucrose (Supplementary Table 1), which was used for all culture experiments in this report. Firstly, we assessed four fatty acids on *R. clarus* asymbiotic culture: potassium myristate (C14:0-K), potassium palmitoleate (C16:1Δ9Z-K), potassium palmitate (C16:0-K) and 2-hydroxytetradecanoic acid (2OH-TDA, which can promote hyphal elongation of *Gigaspora* spp.[30]). The effective concentration of myristate for asymbiotic culture of *R. irregularis* was determined in the previous research[15], thus we adopted a fatty acid concentration of 500 μM for supplying to the medium. Parent spores as a seed fungus collected from in vitro monoxenic culture were placed separately on a solid medium and incubated for 6 weeks. Spore numbers per plate in all experiments in this report are described in Supplementary Data 1. Unlike other tested fatty acids, only potassium myristate activated hyphal growth of *R. clarus* (Fig. 1a and Supplementary Fig. 1). In asymbiotic culture supplemented with potassium myristate, AM fungus expanded its habitat by generating straight, thick hyphae (runner hyphae; RH) with small bunches of short branches, but sporulation rates, the number of secondary spore-forming parent spores per germinated parent spores, were very low 6 weeks after incubation (WAI) in the three trials (Fig. 1b). Similarly, almost no parent spores formed secondary spores by the application of the other fatty acids.

As AM fungi could potentially utilise organic nitrogen for their growth[17–19], we tested the effect of malt extract, SC dropout, yeast extract and peptone on hyphal growth and sporulation of *R. clarus* in the presence of potassium myristate. Among the four, peptone showed higher sporulation rates of 47–100% (Fig. 1c) and the highest production of secondary spores (Fig. 1d and Supplementary Fig. 2), albeit sporulation rates of three trials varied. In the other conditions, parent spores extended their hyphae and produced low numbers of secondary spores, thereafter the fungal growth was arrested. *R. clarus* produced many branched hyphae and densely packed coils (DPCs)[12,13,15], massive assemblies of thin hyphae, in the presence of peptone (Fig. 1a and Supplementary Fig. 2). Although no significant differences were found in the secondary spore production in the media supplemented with any combinations of 100 or 500 μM potassium myristate and 0.2 or 1.0 g l⁻¹ peptone, the maximum values were observed at higher their concentrations (Supplementary Fig. 2), leading us to employ this combination in the following asymbiotic culture experiments. In this survey of the base medium, we observed large variations in sporulation rate and the number of secondary spores among trials. This is probably attributed to the physiological status of each parent spore because spore maturity and degree of spore dormancy may differ among spores produced by in vitro monoxenic culture and the season when experiments were performed (Supplementary Data 1). Accordingly, we performed three trials in each experiment and showed all data in case large variations among trials were observed.

**Strigolactone enables the production of a greater number of secondary spores in asymbiotic culture.** GR24 strongly stimulated the sporulation of *R. clarus* in the presence of potassium myristate and peptone regardless of their concentrations, resulting in the generation of secondary spores in almost all germinated

parent spores at 6 WAI (Fig. 2a). In comparison to the mock treatment, the numbers of secondary spores were greatly increased in a concentration-dependent manner in the presence of GR24 (Fig. 2b and Supplementary Fig. 3). The application of GR24 accelerated the germination of parent spores, which reached over 70%, even at 5 days after incubation (DAI), whereas in the absence of GR24, the rates were less than 40% at 8 DAI and increased subsequently (Fig. 2c). Secondary spores emerged in culture media with GR24 by 8 DAI, in which the sporulation rates were increased to 80% at 14 DAI (Fig. 2d) Whereas only a few germinated parent spores formed secondary spores during the first two weeks of culture without GR24. Consequently, GR24 promoted germination of parent spores and secondary spore formation in *R. clarus*, which finally led to an almost 100% of sporulation rate and the production of a large number of secondary spores. We referred to a base medium containing both 500 μM potassium myristate and 1.0 g l⁻¹ peptone as T medium, and T medium containing 100 nM GR24 as TG medium.

**Jasmonate strengthens the effects of GR24 and increases the number and size of secondary spores in asymbiotic culture.** We tested the effect of MeJA on *R. clarus* asymbiotic culture. In the absence of GR24 (T medium), MeJA exhibited no significant effect on sporulation rate and numbers of secondary spores (Fig. 2e and Supplementary Fig. 4). However, MeJA tended to increase the number of secondary spores in the medium containing GR24 (TG medium). In contrast to GR24, there was no clear correlation between the effect and concentration in MeJA treatment (Supplementary Fig. 4). *R. clarus* finally produced medians of ~300 secondary spores with a maximum of over 700 at 6 WAI in TG medium supplemented with 1 μM MeJA, hereinafter referred to this medium as TGM medium.

Next, we analysed spore size in asymbiotic culture by comparing the size distributions between T, TG and TGM media (Fig. 2f–h). We applied machine-learning-based image analysis to enumerate secondary spores and measure the spore diameter. Spore size had a right-skewed distribution in any medium, with one peak around 40–50 μm in diameter (Fig. 2g). Large spores in the TGM medium appeared brown in colour, similar to parent spores (Fig. 2f). Brown, large spores were also observed in the TG medium, whereas the density was lower than that in the TGM medium. Compared to them, white and small spores were dominant in the T medium. The proportion of large spores, which seemed brown in colour and >70 μm in diameter, was highest in the TGM medium (Fig. 2h).

Additionally, we tested the effects of other phytohormones; jasmonic acid, 1-naphthyl acetic acid, 6-benzylaminopurine, gibberellin A₃ and abscisic acid in the presence of GR24. Jasmonic acid induced the formation of higher numbers and larger size of secondary spores, as did MeJA (Supplementary Fig. 4). *R. clarus* supplemented with 6-benzylaminopurine also tended to produce higher numbers of secondary spores but larger spores were limited (Supplementary Fig. 4). The other tested phytohormones did not show obvious positive effects.

We applied TGM medium to asymbiotic culture of the model AM fungus strain *R. irregularis* DAOM197198. The mean numbers of secondary spores per parent spore were 6–33 spores at 8 WAI (Supplementary Fig. 5), which was 4–22 fold compared with those in asymbiotic culture of *R. irregularis* reported previously[15].

**Time-lapse microscopy reveals development and structure details of *R. clarus* in asymbiotic culture.** To reveal developmental patterns of *R. clarus* under asymbiotic conditions, we followed its growth and sporulation in TGM medium at two-hour

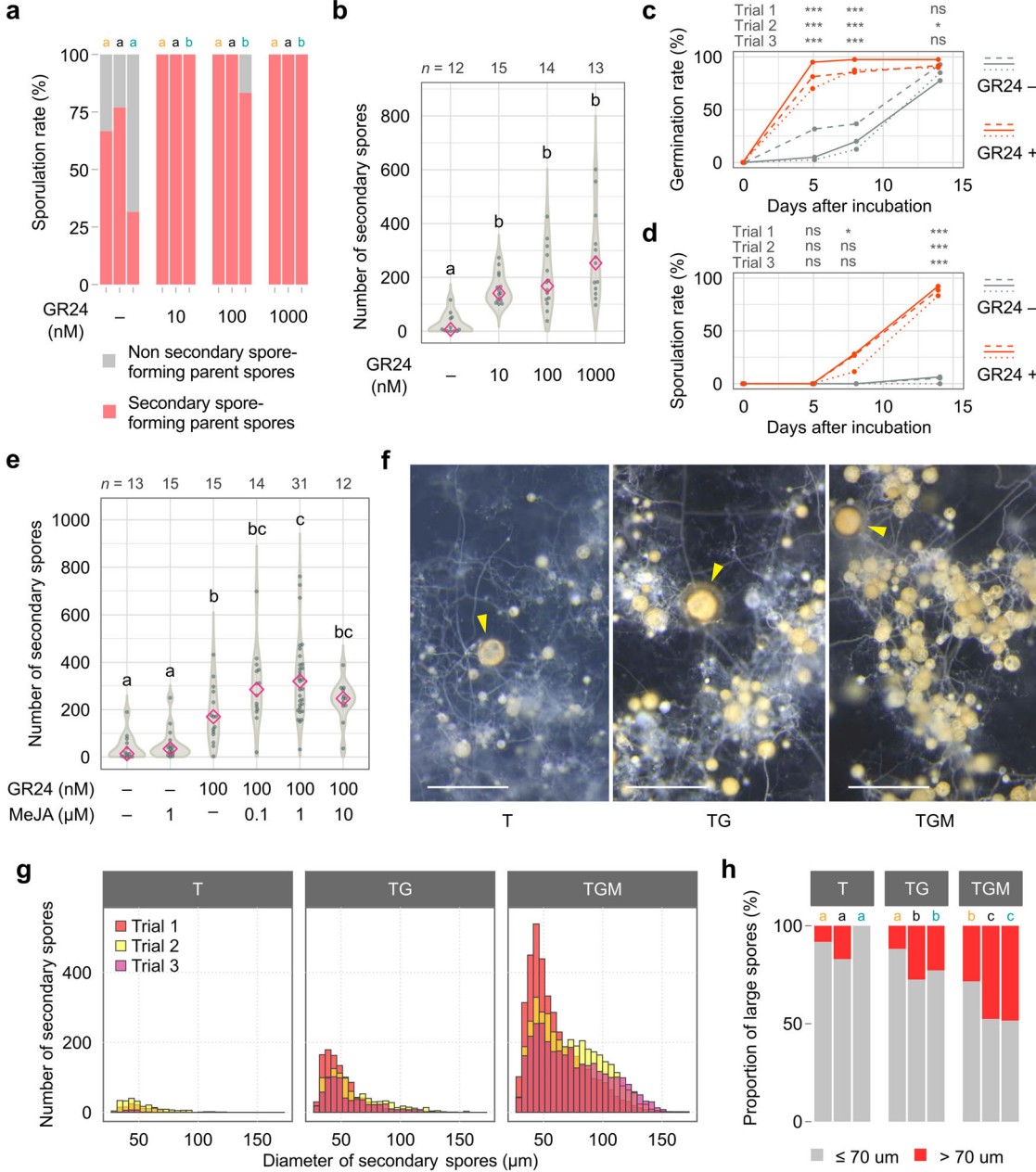

**Fig. 2 Effects of phytohormones on asymbiotic cultures of *R. clarus*.** All culture media contain both 500 μM Myr-K and 1 g l⁻¹ peptone. **a** Percentage of secondary spore-forming parent spores relative to germinated parent spores in cultures with different concentrations of GR24 (10, 100, 1000 nM) at 6 WAI. Three bars in each condition are trials 1 to 3, respectively from the left. Different letters above the graph indicate significant differences among treatments in each trial by Fisher's exact test with Bonferroni correction ($p < 0.05$). **b** Numbers of secondary spores in cultures with different concentrations of GR24 (10, 100, 1000 nM) at 6 WAI. Diamonds indicate medians. Statistical significance was calculated using the Wilcoxon rank-sum test with Bonferroni correction ($p < 0.05$). **c**, **d** Time course of germination rate (**c**) and sporulation rate (**d**) in media with and without 100 nM GR24. Dashed, solid and dotted lines indicate trials 1–3, respectively. Asterisks above graphs indicate significant differences between treatments in each trial using Fisher's exact test with Bonferroni correction. ***$p < 0.001$; *$0.01 \leq p < 0.05$; ns, not significant. DAI days after incubation. **e** Numbers of secondary spores in cultures with different concentrations of methyl jasmonate (MeJA, 0.1, 1, 10 μM) and 100 nM GR24 at 6 WAI. Diamonds indicate medians. Different letters indicate significant differences (Wilcoxon rank-sum test with Bonferroni correction, $p < 0.05$). **f** Secondary spores generated from a single parent spore in T (500 μM Myr-K, 1 g l⁻¹ peptone), TG (500 μM Myr-K, 1 g l⁻¹ peptone and 100 nM GR24) and TGM media (500 μM Myr-K, 1 g l⁻¹ peptone, 100 nM GR24 and 1 μM MeJA) at 8 WAI. Arrowheads indicate parent spores. Large secondary spores were frequently observed in the medium containing both GR24 and MeJA (TGM) than in only GR24 (TG). Many brown spores were observed in TGM and TG media compared to the medium without the phytohormones (T). Bars = 500 μm. **g** Size distribution of secondary spores produced from 8 parent spores in T, TG and TGM media at 8 WAI. **h** Percentage of large spores (>70 μm in diameter) in secondary spores produced in T, TG and TGM media at 8 WAI. Different letters indicate significant differences among treatments in each trial using Fisher's exact test with Bonferroni correction ($p < 0.05$). p-values are described in Supplementary Data 2.

intervals over time for 8 weeks by time-lapse microscopy (Supplementary Movie 1). We summarised its developmental pattern in Supplementary Fig. 6. After the start of culture, parent spores produced germ tubes within one week. Germ tubes began to branch in 4 h after germination in the fastest case and then produced RH. Tree-like branched hyphae were frequently observed from 1 to 2 days after germ tube emergence. These branched hyphae continued to develop and often formed large DPCs. DPC formation proceeded as follows. A single hypha, stemmed from RH laterally, generated short, thin hyphae at short intervals. Thereafter, these short hyphae elongated and branched many times (Supplementary Fig. 6). Large DPC started appearing from one week after germ tube emergence and was formed around parent spores and along with the elongating RH. The fastest secondary spores were formed apically or intercalarily along the lateral branches within 5 days after germ tube emergence. Development of secondary spores finished within 2 days after hyphal swelling (Supplementary Fig. 6). Once formed, these spores did not enlarge anymore during the culture. The final diameter of the spores was positively correlated with the time required for the spore growth after hyphal swelling (Supplementary Fig. 6). *R. clarus* generated only small secondary spores in the early growth stage, thereafter the spore size was increased with days after germ tube emergence (Supplementary Fig. 6).

**Characteristics of asymbiotically generated spores**. We compared asymbiotically generated spores (AS) on TGM medium with symbiotically generated spores (SS) that were produced by in vitro monoxenic culture. A single spore was inoculated into carrot hairy roots on the M medium, then SS were produced on the extraradical hyphae that emerged from the roots. The number of SS in the monoxenic culture was almost the same as that of AS formed in the TGM medium at 8 WAI (Supplementary Fig. 7). Size distribution of SS displayed two peaks (Supplementary Fig. 7). The lower peak around 50 μm was corresponded to the mode length of the diameter in AS, while the higher peak around 150 μm was not seen in any asymbiotic culture, indicating that AS with an intermediate size between the two peaks were increased by the application of GR24 and MeJA (Fig. 2g).

Next, we analysed the composition of fatty acid of triacylglycerol (TAG) in AS in comparison with that in SS. Over 90% of TAG in AS was composed of C14:0 (myristic acid), and only trace amounts of C16:1Δ11 (palmitovaccenic acid) and C16:0 (palmitic acid) were detected. On the other hand, TAG in SS was composed of 67.1 and 32.6% of C16:1Δ11 and C16:0, respectively, and almost no C14:0 was detected. The amount of TAG in SS was over three times that in AS (Supplementary Fig. 8).

**Subculture of secondary spores produced by asymbiotic culture**. To ascertain whether secondary spores of *R. clarus* produced by the first asymbiotic culture can be subcultured, a large single spore (>100 μm in diameter) or a 5 × 5 mm gel block containing 20–40 of large spores (>70 μm) excised from the TGM medium was transferred to a new TGM medium and incubated for 6 weeks. On single-spore subcultures, asymbiotically generated *R. clarus* germinated and extended their hyphae, then formed secondary spores. However, the colony size and the number of secondary spores in single-spore subcultures were smaller than those in initial single-spore cultures (Supplementary Fig. 9). In the gel block culture, no significant differences were observed in sporulation rates and the number of secondary spores per parent spore, compared with the initial single-spore culture (Supplementary Fig. 9). Spores produced in the single-spore subculture were white and smaller than that in the initial asymbiotic culture,

but those in the gel block subculture were large and brown as well as those in the initial culture (Supplementary Fig. 9).

**Asymbiotically generated spores can colonise plants and promote their growth**. We investigated the infectivity and spore productivity of *R. clarus* AS in the in vitro monoxenic culture. A single spore produced in the TGM medium or the one in the monoxenic culture was inoculated into carrot hairy roots grown on the modified M medium. The germination rates of AS and SS were about 85% or higher, and there was no significant difference between the two inoculum sources (Fig. 3a). The rate of daughter spore formation rate was slower in the AS than in the SS up to 4 weeks after inoculation, but no significant difference was observed 4–6 weeks after inoculation. At 8 weeks after inoculation, 82–95% of germinated AS (70–90% of all inoculated AS) produced daughter spores (Fig. 3b, c and Supplementary Data 1). This is a substantial improvement over the rate of 14–67%, in the previous studies[15]. Overall, although AS produced lower numbers of daughter spores compared to SS, some AS had similar spore productivity to SS (Fig. 3d). The numbers of daughter spores were significantly correlated with the diameter of spores inoculated into carrot hairy roots (Fig. 3e).

We also inoculated Welsh onions in pot cultures with SS and AS. The plant growth and intraradical structures in each treatment with the mock, three different numbers of AS (50, 100 and 200 spores) and 50 SS were examined. The effect of inoculation on the growth at 4 WAI tended to be greater in the plants inoculated with 50 SS and 200 AS than in those inoculated with 50 and 100 AS (Supplementary Fig. 10). Furthermore, there were no clear differences in their appearance, shoot dry weight and the intensity of colonisation and arbuscule formation among fungal treatments at 8 weeks after inoculation (Fig. 3f, g and Supplementary Fig. 10).

## Discussion

Jones (1924) reported AM fungi as obligate parasites based on the observation that these fungi eventually die when grown on agar but grow well inside or closely attached to roots[31]. AM fungi have since been recognised as obligate symbionts forming associations with most terrestrial plants[32]. Host-free culture of AM fungi more recently has been developed by supplying fatty acids, myristic acid and palmitoleic acid[14,15]. In this study, we demonstrate that the combination of two phytohormones, strigolactone and jasmonate, facilitates asymbiotical spore production of *R. clarus* in the presence of myristate and the organic nitrogen peptone. The AS can be propagule for the next generation and act as an inoculum that establishes a symbiotic association with plants, although their spore size was still smaller than that of SS.

Myristate and peptone promoted hyphal branching and elongation of *R. clarus*, especially peptone had an important role in sporulation (Fig. 1a–d). The addition of GR24 to them stimulated the early spore formation and probably led to the rapid and more colony expansion (Fig. 2c, d). Since GR24 is a relatively stable compound[33] and is estimated to decrease by 20% per week[34], its effect was thought to continue during the culture and brought about mass production of secondary spores (Fig. 2b). Moreover, GR24 and jasmonates synergistically acted on the increase in numbers and size of secondary spores (Fig. 2e–h). Plant genes involved in jasmonic acid biosynthesis are found to be specifically expressed in arbuscule-containing cortical cells of roots[23,24]. However, the crosstalk between jasmonate and strigolactone signalling in AM symbiosis and the direct effect of jasmonates on AM fungal growth are almost unknown. Future study is required to elucidate their roles in asymbiotic growth of AM fungi.

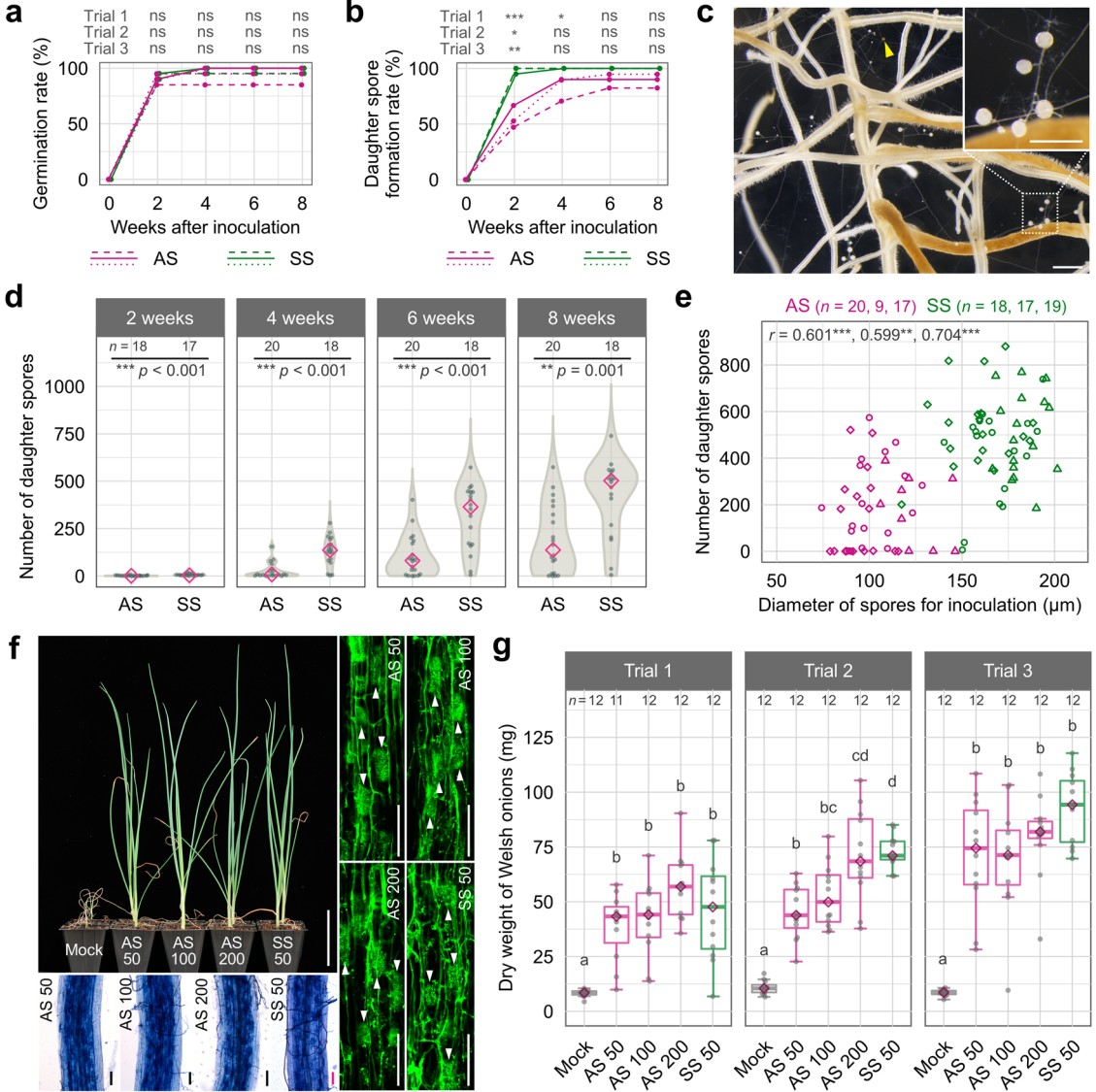

**Fig. 3 Inoculation of asymbiotically generated spores to plants. a–e** Inoculation of asymbiotically generated spores (AS) and symbiotically generated spores (SS) to carrot hairy roots grown on modified M medium. AS and SS of *R. clarus* were prepared by asymbiotic culture in TGM medium and in vitro monoxenic culture, respectively. Time course of germination rate (**a**) and daughter spore formation rate calculated as a percentage of spores that produce daughter spores after the successful colonisation relative to germinated spores (**b**). Dashed, solid and dotted lines indicate trials 1–3, respectively. Asterisks above graphs indicate significant differences between treatments in each time point using Fisher's exact test with Bonferroni correction. ***$p < 0.001$; **$0.001 \le p < 0.01$; *$0.01 \le p < 0.05$; ns not significant. *p*-values are described in Supplementary Data 2. **c** Daughter spore formation in monoxenic culture, in which carrot hairy roots were inoculated with a single (arrowhead). Bars = 500 μm. **d** Time course of daughter spore formation after inoculation of AS or SS. Diamonds indicate medians. Asterisks above graphs indicate significant differences among treatments using Wilcoxon rank-sum test. This is a representative data of all three trials, another two data are described in Supplementary Data 1. **e** Correlation between the diameter of spores inoculated into carrot hairy roots and numbers of daughter spores at 8 weeks after inoculation. Circles, triangles and diamonds indicate trials 1–3, respectively. *r* values are Pearson's correlation coefficient. The three values are trials 1–3, respectively from the left. ***$p < 0.001$; **$0.001 \le p < 0.01$. *p*-values are described in Supplementary Data 2. **f, g** Inoculation of AS and SS to Welsh onions in pots. **f** Growth and root colonisation of Welsh onions inoculated with AM fungi at 8 weeks after inoculation. Numbers following AS or SS are the number of spores for inoculation. The appearance of Welsh onions (upper left, bar = 5 cm). AM fungal structures in roots stained with trypan blue (bottom left, bar = 100 μm) and wheat germ agglutinin conjugated with Oregon Green 488 (right, bars = 100 μm). Arrowheads show arbuscules. **g** Dry weights of Welsh onion shoot at 8 weeks after inoculation. The upper and lower side of boxes show the 25 and 75% quantiles, bars inside boxes indicate medians. Whiskers indicate 1.5 times the interquartile range. Different letters indicate significant differences among treatments in each trial using Wilcoxon rank-sum test with Bonferroni correction, $p < 0.05$.

DPC was first described in co-culture of *R. irregularis* with the bacterium *P. validus*[12] and also in the asymbiotic culture of the fungus supplemented with myristate[15], suggesting that DPC is a specific fungal structure in *Rhizophagus* species cultured without host plants. In *R. irregularis* cultured in TGM medium, relatively large DPCs were formed in the vicinity of a parent spore, and

then small DPCs, identified as branched absorbing structure[35]-like structures in the previous report[15], developed along with RH (Supplementary Fig. 5). In contrast, large DPCs of *R. clarus* were developed not only around parent spores but also away from the spores (Supplementary Movie 1). Fatty acids are absorbed from branched hyphae like DPCs, and part of the fatty acids or lipids

are transferred to newly formed secondary spores via RH in *R. irregularis*[15]. The extensive development of DPCs in *R. clarus* is likely involved in the mass production of large spores, probably by massive uptake of myristate via the structure. In the previous study, *R. irregularis* spores generated asymbiotically could germinate and form new spores in the second asymbiotic culture, but the spore productivity was much lower than that in this study[14]. The higher productivity of *R. clarus* spores shifted the size distribution towards larger spores, enabling us to subculture the fungus at the same level of productivity as in the initial culture.

The final size of AS in *R. clarus*, however, was smaller than that of SS. Spore size may greatly affect the viability/infectivity. The AS produced in this study was capable of colonising plants and promoting their growth, but more spores were necessary to achieve the same effect as that of the SS in the early stage, probably due to their smaller size and the smaller amount of TAG in AS, which could result in limited growth in the pre-symbiotic stage. This idea is supported by Marleau et al.[36] in which the germination rates of four *Glomus* spp. were found to be higher in larger spores with greater numbers of nuclei. *R. clarus* spore development terminated within 2 days after hyphal swelling in our asymbiotic culture conditions (Supplementary Fig. 6), and this is in contrast to the monoxenic culture of *R. irregularis*, in which spores continued to grow for 30–60 days to the mature spore size[36]. It seems likely that the maturation of spores under the asymbiotic conditions might require unidentified plant and/or environmental factors.

This work raises three interesting perspectives for the full understanding and practical use of these important microorganisms. First, we observed some differences in the responses to the TGM medium between the two *Rhizophagus* species (Fig. 3e and Supplementary Fig. 5). Even though *R. irregularis* formed more spores in the TGM medium than in the previous medium in which no phytohormones were supplemented[15], the fungus could produce much fewer spores than *R. clarus*, indicating that the asymbiotic growth of *R. irregularis* might also be stimulated by GR24, but its impact was less than that in *R. clarus*. This would reflect the high interspecific physiological and functional diversity among the fungi[37], probably due to the diversity of Glomeromycotina gene repertoires[38]. Given that the genomes of both *R. irregularis* and *R. clarus* have been sequenced, it would be able to search for lineage-specific missing genes by comparative genome analysis, then lead to establishing custom-made culture methods and identifying key genes, which are involved in the fungal diversity. Second, our culture technique will contribute to biochemical studies. Collecting large amounts of chemical constituents of AM fungi has been difficult due to the nature of the obligate biotrophs. Our technique opens a new window for the analysis of fungal effectors, transporters, small signalling molecules, such as Myc-LCO[39], as well as identification of strigolactone and jasmonate binding proteins via providing an easy and large-scale culture of *R. clarus*. The third contribution involves field studies and the application of fungi to agriculture. The confirmation of the inoculum potential of *R. clarus* AS would be a breakthrough for low-labour-intensive production of the fungi, which will allow extensive field-inoculation studies thus enhancing sustainable intensification of agricultural production via the efficient utilisation of AM fungi.

## Materials and methods

**Fungal materials**. The fungal strain *R. clarus* HR1 was originally isolated from the quarry in Hazu, Aichi, Japan[40] and is available from the NARO Genebank (https://www.gene.affrc.go.jp/index_en.php) as *R. clarus* MAFF520076. Sterilised spores were collected from in vitro monoxenic cultures of *R. clarus* in association with carrot hairy roots[24] grown on M medium[41], solidified with 0.4% of gellan gum (FUJIFILM Wako) instead of agar, at 28 °C in the dark. *R. irregularis*

DAOM197198 (or DAOM181602, another voucher number for the same fungus) was purchased from Premier Tech.

**Preparation of media for asymbiotic culture**. All chemicals were purchased from FUJIFILM Wako unless otherwise noted. Stock solutions of 100 mM potassium myristate, 100 mM potassium palmitate and 10 mM 2OH-TDA (Cayman Chemical) were prepared by dissolving in water and sterilised by filtration. Potassium palmitoleate solution was prepared by adding 10 mM potassium hydroxide to palmitoleic acid (Sigma–Aldrich), then adjusted to 10 mM with water. Bacto™ Peptone (BD), Bacto™ Yeast Extract (BD), Bacto™ Malt Extract (BD) were dissolved in water and autoclaved at 121 °C for 20 min. SYNTHETIC COMPLETE (KAISER) DROP-OUT: COMPLETE (FORMEDIUM) was dissolved in water and sterilised by filtration. Strigolactone GR24 was synthesised as previously described[42] and dissolved in acetone for a 0.1 mM stock solution. MeJA and (−)-jasmonic acid (Sigma–Aldrich) were dissolved in ethanol, and gibberellin A₃ and 6-benzylaminopurine were dissolved in water for a 1 mM stock solution. (S)-(+)-Abscisic acid (Tokyo Chemical Industry) was dissolved in dimethyl sulfoxide for a 5 mM stock solution. 1-Naphthyl acetic acid (a 1 mg ml⁻¹ solution, Sigma–Aldrich) was diluted with water for a 1 mM stock solution. For preparation the base medium described in Supplementary Table 1, all listed compounds except vitamins were dissolved in water, adjusted to pH 6.5 with KOH and sterilised by autoclaving, afterwards vitamins were added. Supplements of fatty acid salts, organic nitrogen sources and phytohormones were added to the base medium under sterile conditions. Stock solutions of fatty acid salts were slowly added with stirring. Media were poured into 90 mm Petri dishes and solidified by cooling.

**Asymbiotic culture**. Parent spores of *R. clarus* as seed fungus for asymbiotic culture were collected from in vitro monoxenic culture, cultivated for at least three months after inoculation (Supplementary Data 1). The spores derived from the same monoxenic culture plates were used for one trial. Spores were separated from extraradical hyphae using forceps and scalpels under an SZX7 stereomicroscope in a laminar flow cabinet. To prevent dryness, collected spores were pooled in sterilised water, then placed separately on a solid medium using a pipette with water droplets under sterile conditions. Large and coloured spores (120–220 μm diameter for *R. clarus* and 80–140 μm for *R. irregularis*) were chosen. Three to ten parent spores of *R. clarus* or *R. irregularis* were placed on a solid medium in a Petri dish (numbers of parent spores are described in Supplementary Data 1). AM fungi were grown at 28 °C in the dark for 6 or 8 weeks. Numbers of secondary spores and spore diameters except those that were analysed by machine-learning-based image analysis (see below) were manually measured under an SZX7 transmitted-light stereomicroscope (Olympus).

**Time-lapse microscopy**. Three or five parent spores of *R. clarus* were placed in the centre of a TGM plate (containing 400 or 500 μM Myr-K). The time-lapse images were captured every 2 h at 28 °C in the dark for 2 months using an SZX7 stereomicroscope equipped with a DP21 digital camera system (Olympus). All images were acquired under constant intensity transmitting light. Time-lapse images were generated by integrating individual digital images with ImageJ[43]. Spore diameters were measured using ImageJ.

**Machine-learning-based image analysis of asymbiotically generated spores**. Machine-learning-based image analysis using Ilastik software (https://www.ilastik.org/index.html) was applied for enumerating secondary spores in asymbiotic culture at 8 WAI. A solid medium containing fungal materials was cut using forceps and scalpels, and then incubated in a three times volume of citrate buffer (1.7 mM citric acid and 8.3 mM trisodium citrate, pH 6.0) for over 2 h at room temperature. After centrifuging at 3200 × g for 15 min at room temperature, the fungal pellets were suspended in 500 μl water and transferred to a new tube. Fungal materials were shredded with an ultrasonic processor (SONICS Vibra-Cell VCX130) as follows: 40% amplitude, 3 sec ON/4 sec OFF pulses for 1–5 min, 2 mm diameter probe. After centrifugation, 300 μl of the supernatant was removed, and fungal pellet was suspended. An aliquot containing ~300 spores was placed on a Petri dish. Spores were dispersed using forceps and allowed to settle down for over 1 h. Digital images of spores were captured using an SZX7 stereomicroscope with a digital camera system (DP73, Olympus) under transmitted light. Spore counting by machine-learning was performed using the pixel classification workflow of Ilastik. First, the training of a classifier that can separate spores from the background was done. Next, individual spore images were extracted by image binarisation using simple segmentation function. The number of spores were counted using the binarised images by ImageJ[43]. Spore diameter was estimated from the maximum Feret diameter measured by ImageJ. Spores smaller than 30 μm in diameter were excluded from the analysis.

**GC-MS analysis of TAG**. AS was cultured in TGM medium at 28 °C in dark for 8 weeks. SS was cultured with carrot hairy roots. Spores were collected by melting gels using citrate buffer as described above. Additionally, the fungal materials in gels were suspended in 30 ml of 1% (v/v) Triton X-100 (Sigma–Aldrich), incubated for 20 min at 50 °C and centrifuged to collect the fungal pellets. They were washed with distilled water and centrifuged three times. The remaining water was

completely removed with filter paper and the fresh weight of the fungal pellet was measured. Chloroform–ethanol (2:1) solution (200 µl) was added to the fungal pellet and sonicated in an ultrasonic water bath for 1.5 h. After centrifugation, the lipid extract solutions were concentrated by nitrogen gas. The concentrates were purified by preparative silica gel TLC (Kieselgel 60 F$_{254}$, Merck) using $n$-hexane–diethyl ether–acetic acid (80:30:1) as a developing solvent to yield TAG. Heptadecanoic acid methyl ester was added as an internal standard to the TAG region on the TLC plate. Fatty acid methyl esters were prepared by incubating the purified TAG in $n$-hexane: 1 M methanolic KOH [10:1 (v/v)] as described previously[44]. An aliquot of the upper hexane layer was applied to GC-MS. GC-MS data were recorded using a GC-MS-QP2010 Plus (Shimadzu) and an InertCap 5MS/NP column (25 m×0.25 mm, 0.25 µm film, GL Sciences). The conditions used were as follows: injection 1 µl (splitless, 60 s valve time), injector temperature 200 °C, carrier gas He (at 0.8 ml min$^{-1}$), transfer line temperature 300 °C, ion source temperature 230 °C, electron energy 70 eV. The temperature of the column oven was programmed as follows: 60 °C for 2 min, followed by an increase to 160 °C at 25 °C min$^{-1}$, and then to 300 °C at 5 °C min$^{-1}$. Identification and quantification of fatty acid methyl esters were performed in the scan mode. Fatty acid composition (C14:0, C16:0 and C16:1Δ11) was determined by calibration with internal standard.

**Subculture of asymbiotically generated spores**. *R. clarus* secondary spores with a diameter of >100 µm generated by asymbiotic culture of at least 3 months were used for subcultures (Supplementary Data 1). A single spore or an agar block was transferred to a new TGM medium using forceps and scalpels. Agar blocks were a 5 mm square in size and contained ~20–40 secondary spores over 70 µm in diameter. AM fungi were incubated at 28 °C in the dark for 6 weeks. Numbers of newly formed secondary spores were counted under an SZX7 stereomicroscope.

**Inoculation test of asymbiotically generated spores to plants**. For the inoculation test of *R. clarus* into carrot hairy roots, spores produced by asymbiotic and in vitro monoxenic culture for at least 3 months were used as inoculum (Supplementary Data 1). *R. clarus* secondary spores with a diameter of >70 µm generated by asymbiotic culture were chosen. A single spore was harvested using forceps and scalpels under an SZX7 stereomicroscope and placed on the vicinity of carrot hairy roots grown on M medium. The production of daughter spores on the extraradical hyphae emerging from hairy roots was observed under the stereomicroscope.

For inoculation to Welsh onion, *R. clarus* spores produced by asymbiotic and in vitro monoxenic culture for at least 4 months were used as inoculum. Secondary spores formed in asymbiotic culture were collected after melting solid medium by citrate buffer as described above. Seeds of Welsh onion cultivar Asagi-Kujo-Hosonegi (TOHOKU SEED, Japan) were sown in sterilised Akadama soil and incubated in the dark for 5–7 days at 24 °C. One day before inoculation, seedlings were transferred to long-day conditions (16 h of light and 8 h of dark). Sterilised soil (1:1:1 mixture of black soil: Akadama soil: river sand) for pot culture was prepared as described by Ohtomo et al.[45]. Briefly, all soil was autoclaved at 100 °C for 60 min, only black soil was sterilised twice and air-dried for a month. Spores were mixed in 40 ml of sterilised soil that was supplemented with 20 ml of a modified Long Ashton liquid medium[46] containing 20 µM phosphate (Supplementary Table 2). One seedling was transplanted in a 4 cm square pot filled with the inoculated soil and cultivated at 28 °C for 8 weeks. After harvesting, plant shoots were dried at 80 °C for 4 days or 50 °C for 2 weeks and weighed with an analytical balance.

**Observation and evaluation of AM fungal structures in roots**. All roots were cleared by 10% (w/v) KOH solution at 70 °C for 2–3 h. For trypan blue staining, samples were washed with water three times, then transferred to 0.05% (w/v) trypan blue in lactic acid and heated at 70 °C for 2–3 h. For wheat germ agglutinin conjugated with Oregon Green 488 (Thermo Fisher Scientific) staining, samples were washed with PBS solution and transferred to 1–2 µg ml$^{-1}$ dye solution and incubated at room temperature for 1 h. Images of trypan blue and wheat germ agglutinin staining were acquired using a DM2500 microscope equipped with a DFC310 FX digital camera system (Leica Microsystems) and a Nikon A1 confocal laser scanning microscope, respectively. The frequency and intensity of mycorrhizal colonisation, and arbuscule abundance were evaluated according to Trouvelot et al.[47], summarised in https://www2.dijon.inrae.fr/mychintec/Protocole/Workshop_Procedures.html. For this assessment, we observed root fragments using an SZX7 stereomicroscope.

**Statistics and reproducibility**. Regarding asymbiotic culture experiments, 6–36 parent spores were incubated in one trial and germinated spores were analysed. To compare AS with SS, 20–21 spores were examined in one trial. To evaluate the growth of Welsh onions, 11–12 plants were analysed in one trial. All experiments were done three times and their tendencies were examined. All statistical analyses were performed by R software (version 4.0.2). The differences among experimental conditions in each trial were tested with the Wilcoxon rank-sum test (for the number of spores, dry weight and mycorrhizal colonisation rates) and Fisher's exact test (for germination/sporulation rates). For multiple comparisons, $p$-values were corrected by the Bonferroni method. Pearson's correlation coefficient was used to analyse the statistical relationship.

**Reporting summary**. Further information on research design is available in the Nature Research Reporting Summary linked to this article.

## Data availability
All source data for figures are included in the Supplementary Data 1.

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

## Acknowledgements

We would like to thank Prof. Masanori Saito for his kind advice. We also thank Yumi Yoshinori, Yuuko Ogawa and Asami Tokairin for experimental support. The modified low-P Long Ashton medium was developed by Dr. Ei-ichi Murakami. Dr. Ryo Ohtomo and Dr. Yoshihiro Kobae kindly advised us about the inoculation test. This research was supported by the Model Plant Research Facility of NIBB by lending the necessary equipment. Confocal images were acquired at Spectrography and Bioimaging Facility, NIBB Core Research Facilities. This work was supported by ACCEL (JPMJAC1403) form the Japan Science and Technology Agency (to T.E., K.S., K.A. and M.K.).

## Author contributions

All AM fungal culture experiments and surveys of fatty acids and phytohormones were performed by S.T. and K.H. Results of culture experiments were analysed by K.H. A survey of organic nitrogen sources was done by H.K. Time-lapse images were acquired by Y.K. and K.H. and analysed by Y.K. Inoculation tests of AM fungal spores into Welsh onions were performed by S.T., K.H. and K.Y. Data were statistically analysed by T.M. and K.H. The synthesis of GR24 and the analysis of the fatty acid composition were performed by K.A. The manuscript was written by K.H., K.S., T.E., Y.K. and M.K. This study was planned by T.E., K.S., K.A. and M.K and supervised by M.K.

## Competing interests

The authors declare no competing interests.
