## [Transparent Peer Review File · Communications Biology]

Reviewers' comments:

Reviewer #1 (Remarks to the Author):

In the manuscript "Asymbiotic mass production of the arbuscular mycorrhizal fungus *Rhizophagus clarus*" the Authors aim to elucidate the hormonal control of the fungal life cycle and developed a new inoculum production protocol.

They evaluated *R. clarus* and *R. irregularis* growth and sporulation in asymbiotic culture using different media. The Authors previously demonstrated that myristate initiated the asymbiotic growth of AM fungi and can also serve as a carbon and energy source. In this MS they investigated how to improve fungal biomass and spore production in the asymbiotic culture systems.

They found that the asymbiotic growth and the sporulation rate of *R. clarus* was stimulated by the presence of potassium myristate, peptone as nitrogen source, and by two phytohormones, (strigolactone and MeJA) which act synergically in the secondary spores production.

Taking advantage of time-lapse microscopy tool, the Authors also reveal the development and structure details of *R. clarus* in asymbiotic culture. Finally, Authors ascertain whether secondary spores of *R. clarus* produced by asymbiotic culture can be subcultured and can successfully colonize Welsh onion roots.

Following the Authors demonstration, a new base medium for asymbiotic culture of *R. clarus* HR1 is now available. This MS offers new and fascinating data and opens new interesting perspectives in the field of AM symbiosis, looking in particular at the fungal partner.

However, in my opinion this MS would need some additional experiments to better explain, at least, some biological issues raised from these data. I think that the Authors should tempt to clarify some issues, such as the role of peptone or MeJA in the asymbiotic phase of AM symbiosis and how this a promising large-scale culture system protocol would affect the functionality of AM symbiosis in planta.

More detailed comments are provided below:

1) As cited in line 67, it is already known that MeJA was increased during AM fungal colonization in roots. However, the role of MeJA has not been assessed during the asymbiotic phase or the early stage of the symbiosis. I am wondering if the Authors could quantify in root tissues and/or root exudates the MeJA during the asymbiotic phase treating the host plants with LCO or CO or GSE. I think that this would help to understand if MeJA is playing an active role in the molecular dialogue between partners.

2) MeJA alone is not altering the *R. clarus* growth and sporulation in asymbiotic condition but it increases the SLs effect. Did the Author check other compounds/phytohormones to test the specificity of MeJA+SLs response?

3) Lines 120. It is not clear to me how many biological replicates and how many independent experiments the Authors performed.

4) Lines 128-129: In the Material and Method section, the Authors claimed that they applied 100nM Gr24 in the media. I suppose that the GR24 was applied only once. Considering the natural and synthetic SL analogues exhibit limited stability in aqueous solutions, I am wondering how one application of GR24 could promote the sporulation until 6WAI.

5) Lines 145-146: in this paragraph I did not see any data referring to spore size of secondary spores upon MeJA treatment. Please check the title of the paragraph or add the data in the paragraph.

6) Line 152: Which is the stability of MeJA in the media? Or in water?

7) Line 187: What does it mean right tail?

8) Line 210-211: I did not figure out the differences between "reduced number of spores" and "high sporulation". Could the Authors please clarify this issue?

9) Lines 224- 228: Could the Authors propose an explanation about the temporary delay between AS and SS to produce daughter spores.

10) Lines 259-261: The Authors claimed that "The rapid colony expansion stimulated by GR24 in TG

or TGM medium may further augment the amount of nutrients absorbed into its hyphae'. I think that the Authors could try to address this issue i.e. measuring the P or N content in AM fungus or in the media after the assay.

11) Lines 296-300: the Authors extended the characterization of AS spores during the symbiotic phase, colonizing with AS and SS Welsh onions root. It is unclear to me if the frequency/intensity of the mycorrhization was evaluated (i.e. by morphological analysis, Trouvelot et al., 1966) in both conditions. I think that these data should be provided.

12) Lines 400-401: how many spores have the Authors evaluated to assess the spore number and size?

13) Lines 597 and Figure 2: different Gr24 and MeJA concentrations have been tested. However, neither in the results nor in the material and method sections I have seen these data commented or the treatment details.

14) Figure 3. The pictures of inked mycorrhizal roots are not clear and they did not add too much without a percentage of colonization data.

Reviewer #2 (Remarks to the Author):

This is an interesting work, showing that the use of two plant hormones, strigolactone and methyl jasmonate, together with myristate and organic nitrogen, induces the production of a large amount of spores in asymbiotic cultures of the arbuscular mycorrhizal (AM) fungus *Rhizophagus clarus* HR1. The Authors showed that such spores were able to promote the growth of onion, consistently with the growth responses obtained by using symbiotically produced spores. These data represent a major step forward in the axenic cultivation of AMF, which have so far been considered non-culturable. Moreover, the findings of this study will boost a less labour-intensive production of high amounts of AMF inoculum, to be utilized in sustainable agroecosystems.

There are some minor faults in the "Perspective" Section.

1) Line 304: The sentence "This study raises three interesting perspectives for this microorganism" should be modified, as AMF are a group of symbiotic fungi, not a single microorganism. It could be "This work raises three interesting perspectives for the full understanding and practical use of these important microorganisms".

2) Lines 306-307: this sentence does not seem grammatically sound.

3) Lines 309-310: The sentence "It implies that pathways involved in AM fungal growth could be different even within the same genus, *Rhizophagus*" should be modified. It has long been known that AMF are very diverse, as regards the growth, biology, biochemistry, functionality, infectivity, efficiency and genetics of each isolate within each species, and even of each lineage within each isolate. Thus, the word "implies" is not appropriate.

4) Line 318: AS is plural, so the verb should be "promote", not "promotes".

5) Lines 391-320: the Authors could mention also that the findings of their work will boost a less labour-intensive production of high amounts of AMF inoculum, to be utilized in sustainable agroecosystems.

6) Lines 321-326: These 6 lines should be deleted, as out of context and unfounded.

Manuela Giovannetti

Reviewer #3 (Remarks to the Author):

Sachiko et al. investigated the effect of two phytohormones, strigolactone and methyl jasmonate on

the growth and the sporulation of *Rhizophagus clarus* HR1 in asymbiotic cultured supplemented with potassium myristate and organic nitrogen. They showed that the combination of these two phytohormones improves asymbiotical spores production of *R. Clarus* in the presence of myristate and peptone. These induced spores are smaller than the spores produced by co-culture with plant and can be subcultured. They have the ability to produce daughter spores after the colonization of carrot hairy roots and to promote the growth of welsh onions after their inoculation. The topic of this paper certainly fits the journal's scope. The statistical methods used are appropriate for the dataset. It makes an interesting contribution to the literature. On the whole, this is a well-presented investigation, worthwhile for publication in communications biology after being revised.

General comments:

1. The introduction is not thorough. The manuscript will benefit by the addition of some sentences about the justification of the use of different organic nitrogen sources.
2. The application of yeast extract showed high sporulation rate (Fig 1C) and low number of secondary spores (Fig 1d). Could you explain this inconsistency?
3. The number of daughter spores is significantly correlated with the diameter of spores (Fig 3e). Do you have a hypothesis to explain this ?
4. The authors showed the AMF structures in roots inoculated with AS and SS but did not show mycorrhization rates. Can authors add this missing information?
5. In perspective, the authors report the interest of biochemical analysis and the difficulty to collect large amounts of chemical constituents of AM fungi. Would it be possible to compare the fatty acid composition of Symbiotically-generated and Asymbiotically-generated spores? In Line 295, the authors explained the lower mycorrhizal effect of AS on plants growth by the small spore size and the delayed establishment of AM symbiosis. It is likely that a low lipid content in AS impact their ability to colonize the roots of a host plant. This point is not discussed ?

Specific comments:

- Line 59 : Add a reference
- Line 89 : Add in the text, the abbreviations of fatty acids used in figures : C14:0-K, ...
- Lines 96-99 : The two sentences could be merged
- Lines 113 : "analyses of phytohormones" : the sentence could be rewritten
- Line 206 : "new medium" : add a precision on the composition of this medium
- Line 211 : "on the medium" could be deleted
- Line 333 : "instead of agar" could be deleted
- Line 587 : justify the choice of these concentrations (500 μ M Myr-K and and 1 mg/l peptone) in the text
- Line 571 : There are no yellow arrows in the rightmost picture.
- Figure 1c : For Mock, the letters are a / a / a : Are you sure ?

Figure 3f and line 635 : what is the number of spores in the SS condition?

- Supplementary Figures 1 & 2 : "on asymbiotic culture" : the title of these figures could be more precise
- Supplementary Figure 2a (text) : Number() of secondary spores /
- Supplementary Figure 2c (text) : spore(s) number / 0.2 or 1 mg L-1
- Supplementary Figure 3 (text) : add different concentrations tested
- Supplementary Figure 9b : These pictures could be deleted
- Supplementary Table 2 : Number of germinated parent spores and germinated rate are two redundant information.

Dear editor and reviewers,

We appreciate your helpful and constructive feedback. We have added the data according to your suggestions, and we have revised the English of the manuscript to make them more comprehensive and readable without changing its meaning. Additionally, we found some mistakes in the manuscript by ourselves. We sincerely apologise and have corrected the parts as following.

The unit of peptone concentration “mg l⁻¹” had been wrong, we have revised all the incorrect units as correct units “g l⁻¹”.

The *p*-values in Supplementary Table 2 had been partially incorrect and we have corrected them. These corrections of *p*-values do not affect the results.

The method of how to calculate the size of spores had been partially wrong and we have rewritten the part in the Materials and Methods section.

Our responses to each reviewers’ comments were listed below.

For referee#1,

We are thankful for a lot of your helpful, constructive and valuable comments, advice and suggestions. We have revised our manuscript and responded to your questions or comments below.

> 1) As cited in line 67, it is already known that MeJA was increased during AM fungal colonization in roots. However, the role of MeJA has not been assessed during the asymbiotic phase or the early stage of the symbiosis. I am wondering if the Authors could quantify in root tissues and/or root exudates the MeJA during the asymbiotic phase treating the host plants with LCO or CO or GSE. I think that this would help to understand if MeJA is playing an active role in the molecular dialogue between partners.

Response: Hause et al. 2002 reported the correlation between mycorrhizal formation and jasmonic acid. They measured the endogenous JA in barley roots which were transplanted into pots containing mycorrhizal leek. The amount of JA started to increase from 10 days after the transplantation, which is the stage after arbuscule formation, and has been increasing at 14 days after transplantation, which is the stage that the colonisation rate started to decrease. Based on the result the authors proposed a model in which the induction of JA biosynthesis in colonised roots is linked to the stronger sink function of mycorrhizal roots (Hause et al. 2002, Plant Physiol.). Judging from the stage where JA biosynthesis is induced, it is considered that some fungal factors work relatively from middle to late stages in the root colonisation in the cortex, rather than the early communications between the root surface and AM fungi. This is a challenging issue for the future, thanks for your comment.

> 2) MeJA alone is not altering the *R. clarus* growth and sporulation in asymbiotic condition but it increases the SLs effect. Did the Author check other compounds/ phytohormones to test the specificity of MeJA+SLs response?

Response: We have examined the effects of other phytohormones on *R. clarus* asymbiotic culture according to your suggestion. We had demonstrated that a sole addition of MeJA did not show a clear effect on sporulation during asymbiotic cultures. Therefore, we assumed that SL is essential for asymbiotic mass production and examined these effects in the presence of SL. We tested five phytohormones; ABA, GA, NAA, BAP and JA. We found that JA showed a similar effect to MeJA, while the other four phytohormones did not show clear positive ones. Considering these results, we think it is suggested that jasmonates have a particularly high effect on the asymbiotic culture. We added the results to the manuscript (L176–182). We think this data has strengthened our argument.

> 3) Lines 120. It is not clear to me how many biological replicates and how many independent experiments the Authors performed.

Response: In all asymbiotic culture experiments, the parent spores that were collected from the same monoxenic culture plates were used in one trial. The numbers of tested parent spores in each experiment were shown in Supplementary Table 2. The results of these experiments, especially in the sporulation rates and the number of secondary spores, varied among trials possibly due to the difference in the physiological status of parent spores. Therefore, we have done three trials in each experiment. We analysed the statistical significances among conditions in each trial by Fisher's exact test (for germination/sporulation rates) or the Wilcoxon rank-sum test (for the number of secondary spores), consequently we found the tendencies among conditions in each experiment. We showed all data in case we observed large variations among trials. We have added description in the results (L133–135) and the Materials and Methods (L387–388, L506–508)

> 4) Lines 128-129: In the Material and Method section, the Authors claimed that they applied 100nM Gr24 in the media. I suppose that the GR24 was applied only once. Considering the natural and synthetic SL analogues exhibit limited stability in aqueous solutions, I am wondering how one application of GR24 could promote the sporulation until 6WAI.

Response: GR24 is a relatively stable compound in strigolactones (Akiyama et al. 2010, Plant Cell Physiol.) and decreases only 20% after 7 days (Umehara et al. 2015, Plant Cell Physiol.). If it decreases at the same pace, 26% of GR24 would remain after 6 weeks. We added 100 nM of GR24 and we think the effect of GR24 will persist even at 6 WAI. We have added the explanation in the Discussion section (L282–284). We think it makes our results more convincing.

> 5) Lines 145-146: in this paragraph I did not see any data referring to spore size of secondary spores upon MeJA treatment. Please check the title of the paragraph or add the data in the paragraph.

Response: We apologise for the incorrect heading in the previous manuscript. We have reorganised the results section in the revised manuscript and confirmed now the heading has become correct.

> 6) Line 152: Which is the stability of MeJA in the media? Or in water?

Response: Since MeJA is frequently used in cell suspension culture as an elicitor for producing secondary metabolites of plants, it is presumed that it is stable to some extent during the culture period (for example, about 1 month). However, as far as we searched for papers that give any data about chemical stability of MeJA in water or any media, we could not find them.

> 7) Line 187: What does it mean right tail?

Response: We had referred to a gently-descending slope in the right side of AS histograms (Fig. 2g) as a tail. We have removed the sentence including the word to prevent any confusion for readers.

> 8) Line 210-211: I did not figure out the differences between “reduced number of spores” and “high sporulation”. Could the Authors please clarify this issue?

Response: We have deleted these complicated words and have modified the English in this section in your question to make more understandable.

> 9) Lines 224- 228: Could the Authors propose an explanation about the temporary delay between AS and SS to produce daughter spores.

Response: The size of spores and the amount of triacylglycerol in AS were less than those in SS (Fig. 2g, Supplementary Fig. 7c and Supplementary Fig. 8b), which could be the cause of the temporary delay. However, the dry weight of Welsh onions which were inoculated with AS approximated to that with SS. We have discussed these in the manuscript (L308–314).

> 10) Lines 259-261: The Authors claimed that “The rapid colony expansion stimulated by GR24 in TG or TGM medium may further augment the amount of nutrients absorbed into its hyphae”. I think that the Authors could try to address this issue i.e. measuring the P or N content in AM fungus or in the media after the assay.

Response: Our sentences might be misleading. We purely supposed that expanded hyphae via GR24 may absorb more nutrients and form more and larger spores. We did not intend to discuss the effect by absorption of specific nutrients such as P and N. We edited the section including the sentence. The amount of each nutrient and the mechanism of how they act in AM fungal growth is also an important point and will be future assignments.

> 11) Lines 296-300: the Authors extended the characterization of AS spores during the symbiotic phase, colonizing with AS and SS Welsh onions root. It is unclear to me if the frequency/intensity of the mycorrhization was evaluated (i.e. by morphological analysis, Trouvelot et al., 1966) in both conditions. I think that these data should be provided.

Response: We have redone inoculation tests using Welsh onions and analysed the mycorrhizal colonisation according to Trouvelot et al. 1986. We have changed two

conditions for plant culture to improve the stability of the result. The soil composition has changed as follows: only black soil -> a mixture of black soil, akadama soil and river sand (volume ratio of 1:1:1) according to Ohtomo et al. 2019 (Microbes and Environment). Costa et al. 2013 (Acta Scientiarum) reported that 25-28 °C is suitable for *R. clarus* monoxenic culture. We currently cultured *R. clarus* at 28 °C in both asymbiotic and monoxenic cultures and confirmed they grow well at the temperature. Therefore, we cultured plants for the tests at the same temperature. In previous experiments, four plants were cultured in a pot. We cultured one plant in a pot to evaluate the individual mycorrhizal colonisation. The number of spores for inoculation and the size of pots have also been changed according to the changes.

We evaluated the mycorrhization of Welsh onion roots and confirmed that there were no differences between AS and SS in all five indicators; F%, M%, m%, A% and a% (Supplementary Fig. 10). The dry weight of Welsh onion shoots had given unstable values in the previous experiment, but it has exhibited a stable result after the changes described above. There was no apparent difference among all dry weights of plants with fungal inoculation (Fig. 3g), which was consistent with the results of the evaluation.

> 12) Lines 400-401: how many spores have the Authors evaluated to assess the spore number and size?

Response: We used 8 parent spores to evaluate the number and size of secondary spores shown in Fig. 2g, h and Supplementary Fig. 7c. Total secondary spores produced from 8 parent spores were analysed and shown in the histograms of size distribution (Fig. 2e and h). The calculated average number of secondary spores was shown in Supplementary Fig. 7c.

> 13) Lines 597 and Figure 2: different Gr24 and MeJA concentrations have been tested. However, neither in the results nor in the material and method sections I have seen these data commented or the treatment details.

Response: We made agar plates supplemented with the different concentrations of each compound, and put parent spores on them then cultured for 6 weeks (see Materials and Methods, L). Each concentration is shown under their graphs (Fig. 2a, e), and also in the legend. The effect of GR24 on asymbiotic culture significantly increased by concentration-dependent manner (Supplementary Fig. 3), but that of MeJA did not correlate to their concentration (Supplementary Fig. 4). Among tested conditions, only the treatment of 1 µM MeJA showed always significantly higher effects (Fig. 2e). We have added the description in the text (L139–144, L162–163).

> 14) Figure 3. The pictures of inked mycorrhizal roots are not clear and they did not add too much without a percentage of colonization data.

Response: We have retaken the pictures of mycorrhizal roots stained by trypan blue (instead of a black ink) using a different microscope (Fig. 3f). We think that the new pictures are clearer than before but may still be limited in resolution. We only intend to show representative examples of high intense colonisation roots in these pictures, and we

show clear fungal structures in pictures taken by WGA staining. We have added the percentages of colonisation data, please see Response 11.

For referee #2 Dr. Manuela Giovannetti,

Response: We thank for your helpful and comments. We have reflected your requests throughout our manuscript, and answered your questions below.

1) Line 304: The sentence “This study raises three interesting perspectives for this microorganism” should be modified, as AMF are a group of symbiotic fungi, not a single microorganism. It could be “This work raises three interesting perspectives for the full understanding and practical use of these important microorganisms”.

Response: We have reflected your suggestion and revised our manuscript (L324–325).

2) Lines 306-307: this sentence does not seem grammatically sound. Than the medium which was used in the previous report and does not contain any phytohormones¹⁵.

Response: Thank you for pointing this out, we have rewritten the sentence (L327–328).

3) Lines 309-310: The sentence “It implies that pathways involved in AM fungal growth could be different even within the same genus, Rhizophagus” should be modified. It has long been known that AMF are very diverse, as regards the growth, biology, biochemistry, functionality, infectivity, efficiency and genetics of each isolate within each species, and even of each lineage within each isolate. Thus, the word “implies” is not appropriate.

Response: We fully agree with you, thank you for your suggestion. We modified throughout the sentences, reflecting your comment (L331–333).

4) Line 318: AS is plural, so the verb should be “promote”, not “promotes”.

Response: Thank you for your fine proofreading. We have rewritten this part for a better presentation with correct English.

5) Lines 319-320: The Authors could mention also that the findings of their work will boost a less labour-intensive production of high amounts of AMF inoculum, to be utilized in sustainable agroecosystems.

Response: Thank you for your important suggestion, we have described it in the perspective section (L343–346).

6) Lines 321-326: These 6 lines should be deleted, as out of context and unfounded.

Response: As you pointed out, these sentences might have digressed from the main context. We have deleted them from the manuscript.

For referee #3,

Thank you very much for providing important suggestions and comments. We have revised our manuscript according to your advice, and we answered your questions below.

> 1. The introduction is not thorough. The manuscript will benefit by the addition of some sentences about the justification of the use of different organic nitrogen sources.

Response: Thank you very much for providing your important insight. We have added the description about organic nitrogen in the introduction (L64–69) and in the results part (L115) with some appropriate references.

> 2. The application of yeast extract showed high sporulation rate (Fig 1C) and low number of secondary spores (Fig 1d). Could you explain this inconsistency?

Response: In *R. clarus* asymbiotic culture, secondary spores are generated after hyphal elongation and branching, as shown in Supplementary Fig. 6a. As the colony expands, it produced more secondary spores. *R. clarus* on the medium supplemented with yeast extract formed some secondary spores at early growth stage but thereafter their growth was arrested. *R. clarus* on the medium supplemented with peptone continued to expand for a longer time, consequently more spores could be produced. We have added the explanation to the text so that readers can imagine the process that how fungi produce secondary spores (L119–122).

> 3. The number of daughter spores is significantly correlated with the diameter of spores (Fig 3e). Do you have a hypothesis to explain this?

Response: Marleau et al. 2011 reported that small spores with fewer nuclei cannot germinate and the germination rate increases in large grown spores with greater nuclei. Large spores can store not only more nuclei but also more lipids, namely they have more potential to germinate and expand their colony, that would lead to the correlation between the spore numbers and diameters. We have mentioned it in the discussion part (L308–314).

> 4. The authors showed the AMF structures in roots inoculated with AS and SS but did not show mycorrhization rates. Can authors add this missing information?

Response: We have redone inoculation tests using Welsh onions and analysed the mycorrhizal colonisation according to Trouvelot et al. 1986. We have changed two conditions for plant culture to improve the stability of the result. The soil composition has changed as follows: only black soil -> a mixture of black soil, akadama soil and river sand (volume ratio of 1:1:1) according to Ohtomo et al. 2019 (Microbes and Environment). Costa et al. 2013 (Acta Scientiarum) reported that 25-28 °C is suitable for *R. clarus* monoxenic culture. We currently cultured *R. clarus* at 28 °C in both asymbiotic and monoxenic cultures and confirmed they grow well at the temperature. Therefore, we cultured plants for the tests at the same temperature. In previous experiments, four plants were cultured in a pot. We cultured one plant in a pot to evaluate the individual

mycorrhizal colonisation. The number of spores for inoculation and the size of pots have also been changed according to the changes.

We evaluated the mycorrhization of Welsh onion roots and confirmed that there were no differences between AS and SS in all five indicators; F%, M%, m%, A% and a% (Supplementary Fig. 10). The dry weight of Welsh onion shoots had given unstable values in the previous experiment, but it has exhibited a stable result after the changes described above. There was no apparent difference among all dry weights of plants with fungal inoculation (Fig. 3g), which was consistent with the results of the evaluation.

> 5. In perspective, the authors report the interest of biochemical analysis and the difficulty to collect large amounts of chemical constituents of AM fungi. Would it be possible to compare the fatty acid composition of Symbiotically-generated and Asymbiotically-generated spores?

In Line 295, the authors explained the lower mycorrhizal effect of AS on plants growth by the small spore size and the delayed establishment of AM symbiosis. It is likely that a low lipid content in AS impact their ability to colonize the roots of a host plant. This point is not discussed?

Response: According to your suggestion, we analysed the composition of fatty acid of triacylglycerol (TAG) in SS and AS (Supplementary Fig. 8). We found that there was a great difference in the composition between SS and AS and that the amount of TAG in AS was only one third of that in SS. We have added these data to the manuscript (L222–227) and discussed about the possibility that the spore size and the amount of TAG could affect to the symbiotic competence (L308–314).

> Specific comments:

Thank you for your fine proofreading, we have corrected our manuscript according to your comments.

> Line 59 : Add a reference

Response: We have added a description and the reference (Srinivasan et al. 2014, J. Appl. Nat. Sci.) which described the number of daughter spores produced in the monoxenic culture with hairy roots (L57–60).

> Line 89 : Add in the text, the abbreviations of fatty acids used in figures : C14:0-K, ...

Response: We have added these abbreviations in the text (L100–101).

> Lines 96-99 : The two sentences could be merged

Response: We have merged the two sentences (L109–113).

> Lines 113 : “analyses of phytohormones” : the sentence could be rewritten

Response: We have removed the phrase “analyses of phytohormones” (L128).

> Line 206 : “new medium” : add a precision on the composition of this medium

Response: We have added the medium name “TGM” in order to prevent misunderstanding (L233).

> Line 211 : “on the medium” could be deleted

> Line 333 : “instead of agar” could be deleted

Response: Thank you for your proofreading, we have deleted them.

> Line 587 : justify the choice of these concentrations (500µM Myr-K and 1 mg/l peptone) in the text

Response: Sugiura et al. 2020 examined the efficient concentration of myristate for *R. irregularis* asymbiotic cultures, and confirmed that 500 µM of Myr-K shows a high effect on it. Therefore, we adopted this concentration in our experiments. Regarding the peptone concentration, we tested the two conditions but actually there was no clear difference in the spore number between high and low concentrations (Supplementary Fig. 2c). But the maximum value among all tested combinations was always observed in the higher combination; 500 µM Myr-K and 1.0 g l⁻¹ peptone in all three replicates. We have added the explanation to justify our choice (L102–104, L124–128).

> Line 571 : There are no yellow arrows in the rightmost picture.

Response: The rightmost picture is a magnified image of the middle one, and the parent spore is out of the rightmost picture.

> Figure 1c : For Mock, the letters are a / a / a : Are you sure ?

Response: We calculated the significance among the values of each trial. Each “a” in the mock corresponds to each condition of the same trial. Please see and compare the letters in the same colour.

> Figure 3f and line 635 : what is the number of spores in the SS condition?

Response: We have added the number of SS to Fig. 3f.

> Supplementary Figures 1 & 2 : “on asymbiotic culture” : the title of these figures could be more precise

Response: We have unified these titles to be the same as Fig. 1 and Fig. 2: “on asymbiotic cultures of *R. clarus*”.

> Supplementary Figure 2a (text) : Number() of secondary spores /

We have removed the “s”.

> Supplementary Figure 2c (text) : spore(s) number / 0.2 or 1 mg L-1

Response: The two words “secondary spore” are adjectives and we have edited the third word “number” to the plural form “numbers”.

> Supplementary Figure 3 (text) : add different concentrations tested

We have added the concentrations in the legend.

> Supplementary Figure 9b : These pictures could be deleted

Response: We have deleted the pictures. But we think that it is better to show the time-course growth before sampling, therefore we added a representative picture instead of it in Supplementary Fig. 10a.

> Supplementary Table 2 : Number of germinated parent spores and germinated rate are two redundant information.

Response: We have deleted the column of germination rate in Supplementary Table 2.

REVIEWERS' COMMENTS:

Reviewer #1 (Remarks to the Author):

In my view the Authors have sufficiently addressed my comments and improved the manuscript accordingly.

This work adds new insight into the hormonal control of the arbuscular mycorrhizal fungal life cycle and the development of inoculum production schemes.

Reviewer #3 (Remarks to the Author):

The authors have responded favorably to all my requests. Therefore, I recommend the publication of this study in Communications Biology.